# Global Overview of Environmental Enrichment Studies: What Has Been Done and Future Directions

**DOI:** 10.3390/ani14111613

**Published:** 2024-05-29

**Authors:** Érica da Silva Bachetti, Layane Yamile Viol, Arleu Barbosa Viana-Junior, Robert John Young, Cristiano Schetini de Azevedo

**Affiliations:** 1Programa de Pós-Graduação em Ecologia, Universidade Federal de São João Del-Rei, São João del Rei 35701-970, Brazil; ericabachettibio@gmail.com (É.d.S.B.); layane.viol@hotmail.com (L.Y.V.); 2Programa de Pós-Graduação em Ecologia e Conservação, Universidade Estadual da Paraíba, Campina Grande 58429-500, Brazil; arleubarbosa@gmail.com; 3School of Sciences, Engineering & Environment, Peel Building, University of Salford, Manchester M5 4WT, UK; r.j.young@salford.ac.uk; 4Laboratório de Zoologia dos Vertebrados, Departamento de Biodiversidade, Evolução e Meio Ambiente, Universidade Federal de Ouro Preto, Ouro Preto 35402-136, Brazil; 5Instituto de Ciências Exatas e Biológicas, Campus Morro do Cruzeiro, s/n, Bauxita, Ouro Preto 35402-136, Brazil

**Keywords:** captive environment, animals under human care, animal welfare, animal management

## Abstract

**Simple Summary:**

The technique of environmental enrichment is used for animals kept under human care and aims to provide adequate environmental stimulation to allow the animal to express the greatest number of normal behaviours, which ultimately can improve the quality of life of animals. Knowing the global panorama of studies on environmental enrichment can help direct future studies in this area of science, as it can point out gaps in knowledge and help improve methodologies and evaluations. Therefore, the aim of this study was to outline the global panorama of studies on environmental enrichment, indicating trends and gaps in the area. Studies were searched for in scientific databases, and information was retrieved from them, such as the species studied, the type of enrichment used, and its effectiveness in improving animal welfare, among other things. The results showed that environmental enrichment studies are growing, but that some animal groups and species are disproportionately more studied than others. Several trends and gaps were pointed out after this overview, as well as directions for future studies.

**Abstract:**

(1) Background: Environmental enrichment (EE) is a management principle aimed at meeting the needs of animals under human care by identifying and providing essential environmental stimuli to contribute to the integrity of their psychological and physiological well-being. Studies on EE have been carried out worldwide, but consolidated information on how it has been used, who it has been used for, how it has been evaluated, and what gaps still exist in the subject is scarce in the scientific literature. This study assessed, employing a systematic review, the global scenario of research into EE in animals kept under human care over the last 17 years, answering the above-mentioned questions. (2) Methods: A search for EE papers was carried out in the Web of Science and Scopus databases from January 2005 to December 2021, resulting in 2002 articles from which information was extracted. (3) Results: Results showed an increase in the number of articles published on EE, especially in farms, but studies in laboratory environments continue to be more frequent. Mammals and birds are the most studied animal groups. Cognitive enrichment is the least utilised by researchers. The number of publications by researchers from countries in the southern hemisphere is low. (4) Conclusions: Although the technique of EE is being widely used, it is still focused on certain groups of animals in certain captive environments and carried out mainly in the northern hemisphere of the planet. Therefore, the gaps pointed out here need to be filled by future studies.

## 1. Introduction

Environmental enrichment (EE) is a management principle aimed at meeting the needs of the animal kept under human care, providing psychological and physiological well-being [1,2,3,4], and identifying and providing environmental stimuli necessary for animal well-being [4,5,6,7]. Various studies have shown that an enriched environment offers greater behavioural opportunities for the animal, making its behavioural repertoire more indicative of good welfare [8,9,10]. As the captive environment is, usually, less complex than the natural one, the provision of environmental enrichment items (structural, food, sensory, cognitive, and social) [4,5,11] allows for an increase in the display of exploratory behaviours, an increase in interactions with other individuals, and an increase in the richness and diversity of natural behaviours, thereby often contributing to a reduction in unwanted behaviours such as abnormal repetitive behaviours [12,13,14].

EE provision is often directly linked to an animal’s well-being, both in physical and psychological aspects [4,5]. Thus, the use of EE offers a means to improve the quality of life of animals kept under human care by increasing their positive affective experiences and allowing them to display normal behaviours [15,16]. Although it was first scientifically discussed in 1925 [17] and in 1950–1970 [18,19], EE began to be applied more frequently only after the 1980s [20]. Publications involving environmental enrichment to improve animal quality of life began to increase considerably after 1985 [21].

Following the empirical studies published from the 1980s onwards, review articles on the effects of environmental enrichment for specific groups of animals were published [22,23,24,25,26,27,28]. However, very few review studies have evaluated the panorama of the use of environmental enrichment globally for all animal groups and all environments where animals were kept under human care [21,29]. These studies, as well as showing a general overview of environmental enrichment studies for animals under human care and their benefits for their well-being, also pointed out gaps in the subject, suggesting that future studies should be aimed at filling these gaps. Some of the gaps pointed out were: the low number of studies evaluating the effects of EE for farm animals; the lack of studies evaluating the efficiency of EE from physiological data; the low number of studies carried out by researchers from the southern hemisphere of the planet; a small number of studies carried out with fish, amphibians and reptiles; the rare use of cognitive enrichment; among the most studied animal groups (mammals and birds), a low species richness was contemplated (most studies were carried out either with parrots (Aves, Psittaciformes), primates (Mammalia, Primates), carnivores (Mammalia, Carnivora) or rodents (Mammalia, Rodentia)); and a low number of studies reporting neutral or negative effects of the EE used [21]. However, no study has been carried out to assess whether the gaps previously pointed out have been filled by new studies, and a new systematic review of the scientific literature is needed for this assessment.

Among the scientometric studies, the systematic review is a method developed to quantitatively evaluate scientific publications on a given topic, guaranteed by their technical-scientific reliability [30]. In this sense, it should be noted that systematic reviews are critical, as they make it possible to identify and systematise work and identify possible gaps on the subject, making it possible to indicate new lines of research to fill discovered knowledge gaps. Therefore, for the present study, a systematic review was carried out of the research scenario on environmental enrichment for animals kept under human care around the world, presenting the current state of research on the subject and verifying whether the gaps previously pointed out have been filled and if new gaps have appeared.

## 2. Materials and Methods

### 2.1. Bibliographic Survey

The bibliographic survey consisted of a search for academic papers on environmental enrichment for animals kept under human care, where the research protocol only defined the inclusion of scientific articles, excluding conference abstracts, opinions, theses and dissertations, book chapters, and books. The search was carried out using two important electronic bibliographic search platforms, Web of Science (WOS) and Scopus, both chosen for their relevance in academia. WOS is widely used by academics all over the world and is considered the most comprehensive multidisciplinary database available for searching scientific articles [31]. The Scopus platform has the largest abstract and citation database of peer-reviewed literature in the world, making it possible to compare journals based on various metrics [32,33]. The platforms have truly international coverage of publications, which is important because national databases are biassed towards certain types of research [33,34].

The search period for the articles included January 2005 to December 2021. The search period was determined by the final year of searches carried out by the study published in 2007 [21]. In the aforementioned study, references on environmental enrichment were evaluated between 1945 and 2004, with the authors pointing out gaps and future directions in this branch of science. All the studies published before 2005 were and are important for improving the welfare of animals kept under human care and directed the research analysed here. The keywords used to search for articles were ‘environment* enrichment AND animal*’. The asterisk symbol (*) and the word “AND” as an intercession were used as Boolean moderators so that any variations of the keywords were considered.

To find and evaluate all the relevant papers, the articles related to the keywords were selected and screened to check the references. All the data were then imported into the Rayyan programme, a free web application developed by the Qatar Computing Research Institute (QCRI), which contributes to systematic, integrative, and scoping reviews in an intelligent way [35]. The Rayyan programme is a manager that has tools that optimise the screening process, allowing the abstract to be viewed, identifying possible duplicates, providing inclusion and exclusion decisions (related and unrelated work directly to the enrichment assessment), and the database from which the articles were taken, as well as plotting graphs showing the entire screening summary [35].

The selection of articles carried out in Rayyan was based on the recommendations for systematic reviews in the PRISMA statement (Principal Reporting Items for Systematic Reviews and Meta-analyses), which is a minimum set of evidence-based items for reporting systematic reviews [36]. It consists, among other things, of a checklist and a flow chart with the total number of references found. The PRISMA checklist should guide the writing of the systematic review and consists of 27 items that ensure the author covers all aspects of the review [37]. The search for EE in the databases generated a total of 8972 articles between 2005 and 2021, but of these, only 2002 met the inclusion criteria defined previously (Figure 1). The included papers can be seen in the Raw Data sheet (link in the Data Availability Statement, at the end of this paper).

Several pieces of information were obtained from each article analysed: year of publication, author(s), title, journal, impact factor, in 2023, of the journal where the work was published; type of climate in the region where the research took place (tropical or temperate), whether the research was an experiment or a review; type of enrichment (food, structural, social, sensory, cognitive) [5], data on the experimental design [sample sizes, type of evaluative measure (behaviour, physiology, both, other); and whether success was reported in the results after applying the technique (EE success was explicit informed in the papers and not determined by the authors of the present paper). Next, information such as the first author’s institution was collected so that a relationship could be made between the countries of publication, thus generating a map of the distribution of study sites. Finally, information on the taxonomic group studied, such as their zoological order and species, and type of environment (zoo, farm, laboratory, aquarium, pets, shelter, other (sanctuaries, private collectors, and wild animal rescue centres)), was collected.

### 2.2. Statistical Analyses

Descriptive statistics were calculated for the data extracted from the scientometric analysis. The results were presented as absolute and relative numbers whenever necessary (some results add up to more articles than the total number of articles included because some studies evaluated more than one species; some results add up to less than the total number of articles because the articles where the information was not available).

Pearson’s correlation was used to test whether the number of articles was associated with year of publication. A General Linear Model (GLM) with a Poisson distribution was built to evaluate whether the number of publications (response variable) was influenced by the year (explanatory variable). Also, a piecewise regression was run to evaluate the break-point, where the number of publications increased significantly.

The Anderson-Darling normality test was carried out for the data on the number of animals used in the different types of environments (zoo, farm, laboratory, etc.). As the data did not show a normal distribution, the Kruskal-Wallis test with Mann-Whitney post-hoc tests was used.

Spearman’s rank correlations were carried out between the year and the number of animals used in environmental enrichment research and between the number of citations and the impact factor in the journal where the article was published (impact factor for the year 2023). For the latter, we calculated an influence rank (IR = total number of articles X impact factor) [21], since the works on environmental enrichment were published in many multi-disciplinary journals (N = 466 journals). A distribution map of the countries that have carried out the most research into EE was constructed using the Infogram 2023^©^ program. All statistical tests were conducted in Minitab 19^®^. Significance was attained when the *p*-value < 0.05.

## 3. Results

The number of publications showed an upward trend until 2021, but no breakpoint was detected; that is, there was no year where the publishing of EE papers increased significantly (Figure 2). Articles on environmental enrichment were published in a wide variety of journals, and the top five were: Applied Animal Behaviour Science (8.86%, N = 178), Behavioural Brain Research (4.73%, N = 95), Animals (4.03%, N = 81), Animal Welfare (3.48%, N = 70), and PlosOne (3.13%, N = 63) (Table 1). A positive, weak correlation was found between the number of citations and the journal’s impact factor (r = 0.35, *p* < 0.001); that is, the most cited articles were those in scientific journals with the highest impact factor.

In terms of article types, 1734 (86.48%) were experimental articles, and 260 (12.97%) were review articles. As for the environment in which the research was carried out, the majority was carried out on animals in terrestrial environments (94.91%, N = 1902), and a few were carried out on aquatic animals (4.79%, N = 96). Of the studies presented, 1507 were carried out in temperate regions (75.16%), and 255 were carried out in tropical regions (12.72%).

Studies on EE were carried out in 58 different countries, the most frequent being: USA (17.28%, N = 333), Brazil (7.21%, N = 139), England (4.46%, N = 86), Spain (4. 31%, N = 83), China (4.00%, N = 77), Germany (3.32%, N = 64), Italy (3.01%, N = 58), Australia (2.65%, N = 51), France (1.92%, N = 37), and Japan (1.92%, N = 37) (Figure 3). The studies carried out in the other countries totalled less than 2% of the total number of articles published.

The greatest number of published studies evaluated the effects of EE on laboratory animals (55.52%, N = 1161), followed by farm animals (11.72%, N = 245), zoos (9.66%, N = 202), aquariums (2.73%, N = 57), shelters (2.25%, N = 47) and pets (0.19%, N = 4). Other types of sites, such as wild animal rescue centres, private breeding grounds, and sanctuaries, made up a total of 81 articles (3.87%).

The most studied taxonomic group in aquariums was mammals, with the dolphin species *Tursiops truncatus* (15.79%, N = 9), of the order Cetartiodactyla, being the most studied (Figure 4). In the zoo environment, species from the Primates order, particularly the rhesus monkey *Macaca mulata*, gorilla *Gorilla gorilla* (both with 4.95%, N = 10), and chimpanzee *Pan troglodytes* (3.96%, N = 8), were extensively studied, followed by the Carnivora order, particularly lions *Panthera leo* (3.96%, N = 8) (Figure 4). In the farm environment, the most studied order was the Cetartiodactyla (Mammalia), with the species of domestic pig *Sus scofra* being the most frequent (56.73%, N = 139), followed by the Galliformes (Aves) order, where the most frequent species was farm-raised domestic chickens (*Gallus gallus domesticus*) (6.29%, N = 139) (Figure 4). The most studied taxonomic group in the laboratories was mammals, with the order Rodentia being the most frequent, followed by the species of common rat *Rattus novergicus* (49.44%, N = 574) and mouse *Mus musculus* (39.79%, N = 462) (Figure 4). In the shelter environment, mammals also featured prominently, especially the order Carnivora, represented by the domestic dog species *Canis lupus familiaris* (55.32%, N = 26) (Figure 4). On the other hand, research with pets focused mainly on studies with domestic cats. *Felis silvestres cattus* (75%, N = 3) (Figure 4).

Studies with the taxonomic group Mammalia were predominant (79.91%, N = 1671). This was followed by birds (6.55%, N = 137) (Figure 5). Among the invertebrates, the most studied group was the phylum Arthropoda, with nine studies (0.81%), followed by the phylum Mollusca (0.14%, N = 3), and the phylum Cnidaria, which was studied only once (0.05%) (Figure 5).

Concerning the orders of each taxonomic group, for Mammalia, most of the studies were carried out with individuals from the order Rodentia (51.65%, N = 1080), followed by studies with individuals from the orders Cetartiodactyla (10.66%, N = 223), Carnivora (8.70%, N = 182), and Primates (6.46%, N = 135). Within the bird group, the Galliformes order stood out (3.35%, N = 68), followed by the Psittaciformes order (2.20%, N = 46). The other orders were studied in less than 2% of the studies (Figure 6).

Mammals and birds were the groups that received the most EE, especially of the structural and sensory types (and when the two were associated) (Table 2). The groups that received the least EE were invertebrates and amphibians, but, like birds and mammals, the EE items most applied to invertebrates and amphibians were structural and those where two types of EE were associated (Table 2).

The average number of animals used for scientific research was highest for farm animals (mean ± SE: 6383.00 ± 2948.00), followed by pets (mean ± SE: 1086.00 ± 1053.00), aquariums (mean ± SE: 930.60 ± 860.00), others (mean ± SE: 139.20 ± 61.09), shelters (mean ± SE: 96.84 ± 55.12), laboratories (mean ± SE: 66.00 ± 2.99), and zoos (mean ± SE: 8.34 ± 1.05). The number of animals evaluated in studies of farm animals was significantly higher than the number of animals used in studies of other sites, with studies of zoo animals evaluating the fewest animals (Table 3).

There was no significant correlation between the year and the number of animals used in environmental enrichment research (r = 0.006, *p* = 0.84); that is, there has been no increase in the number of animals used in EE research over the 17 years analysed.

The type of enrichment most used by researchers was combined when two types of enrichment were applied simultaneously (39.33%, N = 625; structural and sensory were the most frequent: 447 studies), followed by structural (28.89%, N = 459), sensory (39.33%, N = 625) and food (6.61%, N = 105). The types of enrichment used in each research environment are shown in Table 4.

The benefits of the EE items offered to the animals were predominantly assessed using behavioural data (38.60%, N = 807), followed by a mixture of behavioural and physiological assessment together (26.45%, N = 553), and finally by physiological assessment alone (21.52%, N = 450). Studies that did not clearly state the method of assessment totaled 281 articles (13.44%).

Most researchers reported that environmental enrichment generated positive results (79.10%, N = 1586); 4.14% (N = 83) of researchers reported neutral effects of environmental enrichment on the animals, and only 2.79% (N = 56) reported negative effects of environmental enrichment. In 277 studies, authors did not report if the EE was successful, neutral, or unsuccessful (13.97%). The success attributed to EE was greater when the items were presented in association, with four or more categories together achieving 100% success (Table 5). The cognitive items were the ones that resulted in the most ambiguous outcomes (Table 5).

## 4. Discussion

In summary, there has been an upward trend in the number of studies in recent decades, especially since 2010. The number of studies with farm animals has increased, slightly surpassing the number of studies with zoo animals, but remains well below the number of studies carried out with laboratory animals. Evaluation of the success of EE is still predominantly based on behavioural analyses, but there has been an increase in the number of studies using physiology or behaviour associated with physiology in recent years. Cognitive enrichment is the least applied, as is social enrichment. Concerning the diversification of the animals studied, there is a predominance of rodents, carnivores, primates, and psittacines, but some other orders, such as galliformes and cetartiodactyls, were used in significant numbers of studies. Lastly, the number of EE studies led by researchers from the northern hemisphere is higher than those led by researchers from the southern hemisphere; despite this, Brazil is a positive outlier.

Historically, farmed animals received little attention regarding their welfare, with few EE studies, and economic restrictions and a lack of public pressure are cited as possible reasons for the lack of research on farm animals [21]. The increase in research in this environment in recent years may reflect a redirection by researchers to fill this knowledge gap or a market need since welfare certification for these animals allows products to be sold for higher prices [38], as well as an improvement in production with increased animal welfare [39,40].

Studies focusing on laboratory animals were abundant, showing that the concern to improve the welfare of these animals is great since they are often used in biomedical research and the results of such research need to be reliable [41,42,43]. Some studies have already shown that low levels of welfare in these animals can generate dubious results in these studies [44,45]. The number of sample individuals used in both farm and laboratory environments is notably higher than in other environments, such as zoos, aquariums, shelters, and pets. It is estimated that there are 192 million animals used in labs around the world each year [46], compared to billions of farm animals, a few million zoo animals, and hundreds of millions of pet mammals. It is worth noting that small samples make analyses susceptible to errors, and the sample size is fundamental for validating research [47]. It is important to mention that perhaps the low sample size of studies carried out in zoos and aquariums is related to the rarity of the species kept by the institutions. Also, laboratory studies normally use the minimum statistically valid sample size based on Power Statistics [48], while animals kept in zoos, aquariums, shelters, and as pets may not be considered research subjects in the same way as laboratories do (ex.: laboratory animals are normally euthanased at the end of the experiment, while animals in other environments not).

There are very few studies in zoos compared to laboratories, but the data shows that research in this environment is constantly growing. This can be attributed to the fact that today zoos are directly related to the conservation of species, especially those threatened with extinction [49]. Promoting environmental education is also one of the objectives to be achieved by zoos, and to do this, it is essential to guarantee a high quality of life for the animals, since for the educational message to be conveyed, they need to be in good physical and behavioural condition [50].

The success of EE for all the groups evaluated is notable, especially when different types of enrichment are combined. The number of studies reporting negative or neutral effects was very low. The difficulty of publishing negative or neutral research is widely known and recognised [51,52]. Other studies also showed the success of the EE technique for animals kept under human care in improving animal welfare by reducing the stress they are exposed to [53,54]. These results may indicate the real efficiency of EE in improving animal welfare or the lack of publications on the negative or neutral effects of this technique.

Although the success of the application of the EE technique for animals kept under human care is more pronounced, some studies report the failure of the technique. Some authors emphasise that EE can generate results contrary to those proposed, resulting in low levels of animal welfare [7]. Some researchers suggest that the experimental design may influence the inefficient or ambiguous responses of the EE, especially when the EE items are used simultaneously [47]. This is because it would be difficult to separate the influence of each item on the animals, making the observation of the negative influences of any item complicated to carry out and, consequently, to publish [47].

Concerning methods for assessing the efficiency of EE, there was a predominance of behavioural assessment, with an increase in physiological assessment in recent years, although this assessment is still much lower than the previous one. The lack of physiological assessments of animals under human care can be attributed to the validation of measurement techniques and equipment, the lack of certified laboratory infrastructure, and the high cost of hormonal analyses per sample [40]. However, the number of studies measuring the endocrine stress response [55], as well as those using both assessments (behaviour and physiology), has been increasing in recent years (from 8.55% and 2.80% of the studies pointed out in 2007 [21] to 21.52% and 26.45% pointed out in the present study). As the assessment of behavioural and physiological parameters together contributes positively to a better assessment of animal welfare [54], these measures must continue to be applied in future studies. Since the application of EE advocates an improvement in the animal’s quality of life, it is important to consider the application of non-invasive methods for physiological assessment to avoid acute stress and ensure the animal’s well-being [56,57].

Of the types of EE, combined enrichment was the most used, especially the structural-sensory combination. Structural enrichment combined with sensory enrichment was common, probably because most of the animals evaluated were laboratory rodents. Laboratories often house tens of thousands of rodents, so structural enrichment is the most practical to implement as it has minimal labour requirements compared to other types of enrichment [5]. The choice of enrichment technique will depend on the behavioural characteristics and demands of each species. For example, our study showed that structural enrichment has been the most widely used for farm animals, such as domestic pigs (*Sus scofra*). Structural or physical enrichment, which consists of introducing objects into the enclosure, such as hiding places, sprinklers, logs, and brushes, has been widely used in pig facilities, especially the provision of straw for pregnant sows, a technique used to reduce unwanted behaviour and increase the frequency of lactation [58,59]. Structural items, as previously mentioned, are the most practical to implement due to their low labour requirements [5] and, if thought through correctly, can provide high levels of welfare for confined animals, increasing production [60]. For entrepreneurs, improving welfare and the associated increase in production fulfil both their expectations, increasing profits, and the expectations of the consumer market, which is currently more concerned about the quality of life of animals [39,61].

Food enrichment is widely used in zoos, where the usual diet is offered in an unusual way to the animal (at different times, chopped or whole, hidden or in the feeder, scattered or grouped, etc.) [3,62]. In this way, the animals are stimulated to forage and work to get their food, allowing them to exhibit behaviours that are closer to natural, which improves their well-being [6,63,64,65]. Sensory enrichment, especially sound and olfactory enrichment, is more commonly used in shelter environments, probably due to the ease with which it can be provided, the great olfactory and auditory capacity of dogs and cats, as well as the characteristics of enclosures, which are usually small, with a high turnover of animals and a high need for cleaning [66,67,68].

The number of articles reporting only the use of cognitive enrichment has decreased over the last 17 years (from 3.49% in 2007 [21] to 2.20% in the present study). This result may have been due to the difficulty and high costs of designing and implementing complex cognitive structures for the animals, the wrong categorisation of the item offered to the animals by the researchers, or the lack of interest or time in offering such items (people might think their species is not very intelligent) [69]. In the present study, it was observed that cognitive items were used more for laboratory animals, such as rodents and primates. Cognitive EE items are those that involve cognitive skills developed by offering opportunities to solve problems and control some aspect of the environment and are correlated with one or more validated measures of well-being [70]. In laboratories, many toys that are offered to rodents and primates are considered cognitive items by researchers [71], although at other times they are considered sensory or structural items [2,69]. This difficulty in standardising items is because they can be classified into more than one category. It is therefore likely that the low number of studies reporting the use of cognitive enrichment is underestimated. In any case, the small number of studies evaluating cognitive items still leaves this knowledge gap unfulfilled.

Even though the number of studies on EE has increased over the years, there is still a low diversity of animal groups studied, with mammals and birds being the predominant groups. The mammal group continued to be the most studied, but it showed a drop over the years (from more than 90% in 2007 to 70% now), and this result reflects the relationship between the animal groups and the study environment. In the present study, laboratory and farm animals were the ones that received the most EE, and, in these places, rodents, primates, carnivores, and cetartiodactyls were the most present. Mammals tend to be more charismatic species for humans [72] and larger than most specimens from other groups, which increases their charisma [73]. In addition, they are usually available in greater numbers for study because they are present in laboratories [46], and many are used in the production of food [74]. Finally, they are species that are phylogenetically closer to humans [75,76]. It is therefore more interesting for researchers to apply their efforts to studies with animals from this group. Furthermore, within this group, the species evaluated are also not very diverse, showing that the studies have a large species-specific bias (the studies are mainly carried out with rats and mice, monkeys, pigs, dogs and cats, parrots, and chickens). It is suggested that the effects of different types of environmental enrichment be tested for fish, amphibians, reptiles, and invertebrates, whose studies are practically incipient, as well as birds and mammals from less studied groups.

Finally, regarding the participation of researchers from the southern hemisphere of the planet, in the last decade, the scenario has changed somewhat, with Brazilian researchers standing out, as Brazil has taken second place in the ranking of publications of studies on environmental enrichment, behind only the United States. The greater number of EE studies carried out in Brazil may be related to the increase in animal welfare researchers in the country [77,78], as well as an increase in investment by Brazilian governmental and private agencies in funding this type of study or an increase in people’s perception of the importance of offering a higher quality of life for animals kept under human care [26,79]. However, the number of researchers from the southern hemisphere is still much lower than those from the northern hemisphere. This result shows that researchers in the global south should continue to increase their research on the subject, preferably by diversifying the countries and animals studied. It is also important to discuss that there may be studies that were not included in the search, even though they are related to studies on environmental enrichment. This may have been due to the lack of use of the words enrichment and environmental in the titles or keywords of the articles. To try to reduce this problem, we used boolean moderators and checked all the reference lists of the included articles. Even so, articles may have been missed in the search, which could change the scenario of some of the analyses presented here, like the participation of researchers in the southern and northern parts of the globe. Therefore, a suggestion for future studies would be to include new keywords in the searches and evaluate the grey literature on the subject, with a view to adding more studies to the analyses.

## 5. Conclusions

In conclusion, the importance of applying the EE technique to maintain or increase animal welfare is still growing in all environments where animals are kept under human care. The gaps pointed out in the scientific literature have not been filled, but progress has been made in some of them. Research involving animal species that have been less widely evaluated is recommended, as is an increase in studies in zoos, aquariums, shelters, and domestic environments. Studies evaluating the physiological effects of EE are recommended, especially those that combine physiological and behavioural measures. Cognitive enrichment should therefore be encouraged. Finally, more studies in the southern hemisphere should be encouraged, and partnerships between researchers from both hemispheres could be an interesting alternative. One suggestion would be to evaluate whether Northern Hemisphere studies are relevant to the Southern Hemisphere, but when it comes to EE studies, even if they are applied to different species, the ideas can be adapted for animals kept in the southern part of the world.

## Figures and Tables

**Figure 1 animals-14-01613-f001:**
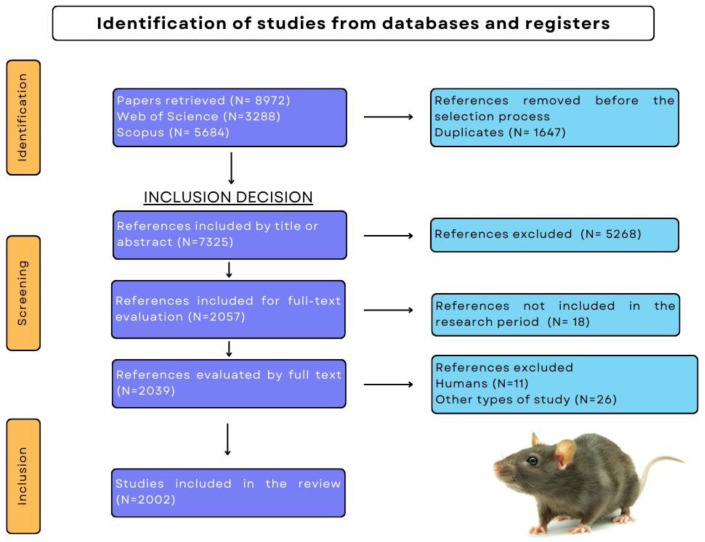
Flowchart depicting the PRISMA protocol for Environment Enrichment research papers reported between 2005 and 2021 in the Web of Science and Scopus databases.

**Figure 2 animals-14-01613-f002:**
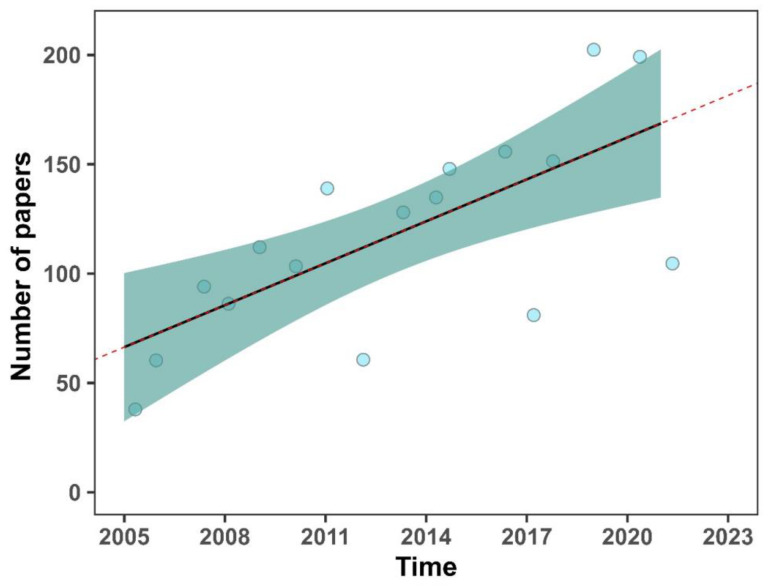
Number of articles on environmental enrichment published between 2005 and 2021. The dotted line represents the trend in the number of publications over time; the solid line represents the regression model showing the temporal breakpoint, where the number of published papers increased significantly. The green shadows show the confidence intervals. Dots represent the number of papers published by year.

**Figure 3 animals-14-01613-f003:**
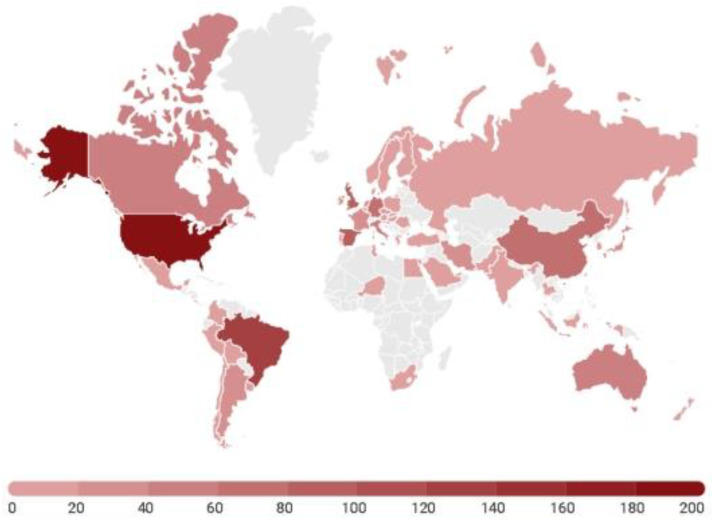
Countries where the environmental enrichment studies were carried out between 2005 and 2021.

**Figure 4 animals-14-01613-f004:**
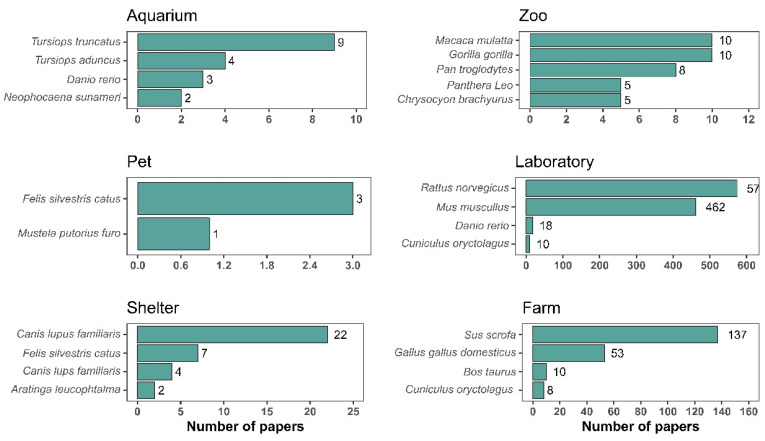
The number of articles and the most frequent animal species in terms of study environment for our selected articles on environment enrichment from 2005 to 2021.

**Figure 5 animals-14-01613-f005:**
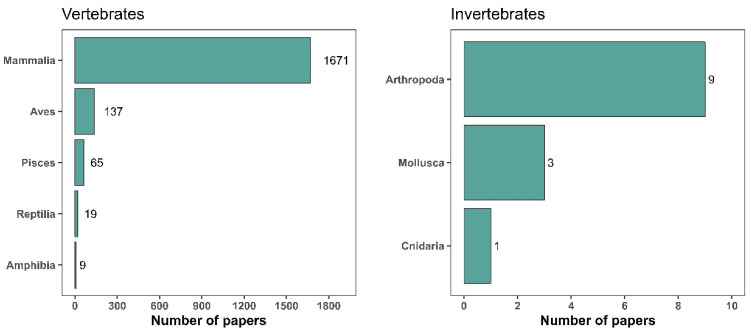
Number of environment enrichment publications by taxonomic group analysed in this study (Vertebrate Classes and invertebrate Phyla), which were published between 2005 and 2021.

**Figure 6 animals-14-01613-f006:**
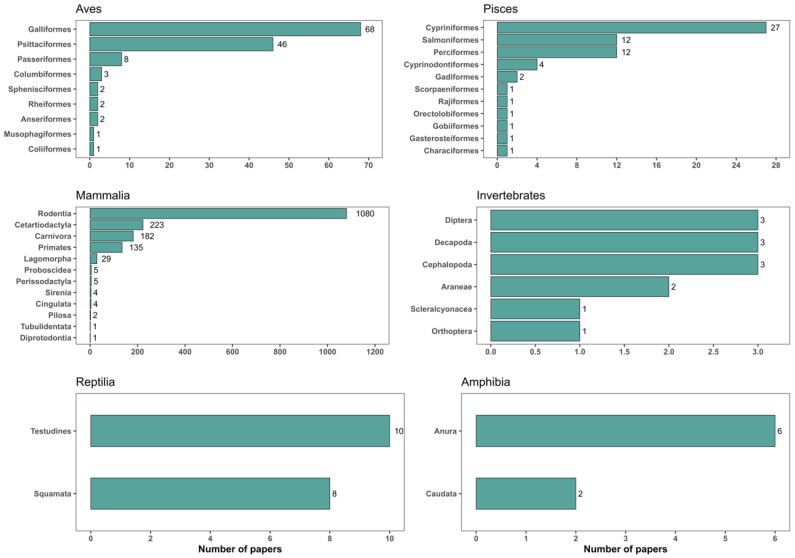
Number of environmental enrichment studies separated by the orders of the major animal taxonomic groups analysed in this study, which were published between 2005 and 2021.

**Table 1 animals-14-01613-t001:** Number and percentage of articles published on EE between 2005 and 2021 in the top 11 journals. Number of articles per journal (N); percentage of the total dataset of articles (%); influence rank (IR) for environmental enrichment (IR = IF × total).

Journal	IF ^a^	N	%	IR
Applied Animal Behaviour Science	2.448	178	8.86	1 (435.74)
Behavioural Brain Research	3.332	95	4.73	3 (316.54)
Animals	2.752	81	4.03	5 (222.91)
Animal Welfare	2.293	70	3.48	7 (160.51)
PlosOne	3.752	63	3.13	4 (236.37)
Neuroscience	7.708	52	2.59	2 (400.81)
Zoo Biology	1.421	51	2.54	10 (72.47)
Journal of Applied Animal Welfare	1.440	46	2.29	11 (66.24)
Physiology and Behaviour	3.742	38	1.89	8 (142.19)
Frontiers in Behavioural Neuroscience	3.617	36	1.79	9 (130.21)
Scientific Reports	4.997	34	1.69	6 (169.89)
Total	3.332 ^b^	744	37.02	

IR = rank position (IR total). ^a^: Impact factor of 2023 (most recent evaluation). ^b^: Median IF.

**Table 2 animals-14-01613-t002:** Types of environmental enrichment offered to different animal taxonomic groups analysed in articles in this study, which were published between 2005 and 2021. Number of studies (%). ND = not informed.

	Invertebrates	Fish	Amphibian	Reptile	Bird	Mammal
Cognitive	-	-	-	1 (5.26)	1 (0.73)	28 (1.68)
Food	1 (7.69)	3 (4.62)	1 (11.11)	1 (5.26)	15 (10.95)	117 (7.00)
Social	-	5 (7.69)	-	-	3 (2.19)	67 (4.01)
Structural	7 (53.84)	35 (53.84)	3 (33.33)	3 (15.79)	41 (29.93)	374 (22.37)
Sensory	-	9 (13.84)	1 (11.11)	1 (5.26)	18 (13.14)	156 (9.34)
ND	2 (15.38)	7 (10.77)	4 (44.45)	4 (21.05)	22 (16.06)	228 (13.63)
Two types	3 (23.09)	4 (6.16)	-	8 (42.12)	27 (19.70)	587 (35.13)
Three types	-	2 (3.08)	-	1 (5.26)	6 (4.38)	95 (5.70)
Four + types	-	-	-	-	4 (2.92)	19 (1.14)

**Table 3 animals-14-01613-t003:** Comparison between the number of animals used in environmental enrichment studies according to the different types of environments analysed in the article in this study, which was published between 2005 and 2021. *** ≤0.001; ns = not significant. N = number of studies.

Environment	N	Median	Kruskal-Wallis	*p*-Value	Post-Hoc Mann-Whitney		
1–Aquarium	29	28.0	380.93	<0.001	1 × 2 ***	2 × 4 ***	3 × 7 ***
2–Farm	213	96.0			1 × 3 ns	2 × 5 ns	4 × 5 ns
3–Laboratory	795	45.0			1 × 4 ns	2 × 6 ***	4 × 6 ns
4–Other	72	14.5			1 × 5 ns	2 × 7 ***	4 × 7 ***
5–Pet	3	47.0			1 × 6 ns	3 × 4 ***	5 × 6 ns
6–Shelter	43	30.0			1 × 7 ***	3 × 5 ns	5 × 7 ***
7–Zoo	143	4.0			2 × 3 ***	3 × 6 ***	6 × 7 ***

**Table 4 animals-14-01613-t004:** Types of enrichment offered to the animals in the different study environments analysed in articles in this study, which were published between 2005 and 2021. Number of studies (%). ND = not informed.

	Aquarium	Farm	Laboratory	Other	Pet	Shelter	Zoo
Cognitive	2 (3.51)	10 (4.12)	9 (0.78)	1 (1.23)	-	2 (4.08)	6 (3.02)
Food	5 (8.77)	15 (6.17)	17 (1.47)	16 (19.76)	-	4 (8.16)	76 (38.19)
Social	4 (7.02)	5 (2.06)	49 (4.24)	2 (2.47)	-	5 (10.21)	7 (3.52)
Structural	15 (26.32)	103 (42.38)	286 (24.72)	20 (24.69)	-	7 (14.29)	17 (8.54)
Sensory	6 (10.53)	21 (8.64)	78 (6.74)	13 (16.05)	1 (25.00)	16 (32.65)	41 (20.60)
ND	10 (17.54)	26 (10.70)	139 (12.01)	4 (4.94)	1 (25.00)	4 (8.16)	16 (8.04)
Two types	10 (17.54)	52 (21.40)	501 (43.30)	17 (20.99)	1 (25.00)	7 (14.29)	25 (12.57)
Three types	3 (5.26)	10 (4.12)	69 (5.96)	5 (6.17)	1 (25.00)	3 (6.12)	9 (4.52)
Four + types	2 (3.51)	1 (0.41)	9 (0.78)	3 (3.70)	-	1 (2.04)	2 (1.00)

**Table 5 animals-14-01613-t005:** The number of studies that reported success (Yes), failure (No), neutrality (Neutral), or that generated ambiguous results (Equivocal) from the use of environmental enrichment, separated according to the type of environmental enrichment analysed in the articles, which were published between 2005 and 2021.

	Yes (%)	No (%)	Neutral (%)	Equivocal (%)
Cognitive	28 (75.68)	-	1 (2.70)	8 (21.62)
Food	132 (95.65)	3 (2.18)	2 (1.45)	1 (0.72)
Social	66 (86.84)	3 (3.95)	4 (5.26)	3 (3.95)
Structural	409 (87.58)	23 (4.93)	30 (6.42)	5 (1.07)
Sensory	163 (87.17)	7 (3.74)	3 (1.60)	14 (7.49)
Two types	555 (91.74)	12 (1.98)	33 (5.45)	5 (0.83)
Three types	100 (91.74)	1 (0.92)	5 (4.59)	3 (2.75)
Four + types	19 (100.00)	-	-	-

## Data Availability

The original data presented in the study are openly available in the Mendeley Data Repository at https://doi.org/10.17632/nd2bhknz5t.1 (accessed on 10 April 2024).

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
