# Peer review of "Global Overview of Environmental Enrichment Studies: What Has Been Done and Future Directions"

_animals, 2024, doi:10.3390/ani14111613_

Round 1

Reviewer 1 Report

Comments and Suggestions for Authors

Very good for a environmental enrichment studies information report. Looking back on the past, the future directions author rarely talk about, but also respect to the readers.

Because the author specifically talked about Brazil, I have to say that in China and Japan, there are so many Chinese and Japanese magazines, which are not included in the English database. In fact, the research is very colorful.

For example: Applied Animal Behaviour Science 168 (2015) 1823.
Asian-Australasian Journal of Animal Sciences (AJAS) 2017; 30(2):
267-274.

Published, perhaps not included, there are many studies related to EE in the Chinese and Japanese literature that suggest removing Fig3 and distinguishing the southern hemisphere from the northern hemisphere means little for talking about "Global overview"

Author Response

REVIEWER 1

Very good for a environmental enrichment studies information report. Looking back on the past, the future directions author rarely talk about, but also respect to the readers.

Response: thank you for your comment. We talked about future directions in the discussion and in the conclusion sections. The suggestions were made based on the gaps found with our data. Thus, we suggested future studies aiming in fulfilling these gaps.

Because the author specifically talked about Brazil, I have to say that in China and Japan, there are so many Chinese and Japanese magazines, which are not included in the English database. In fact, the research is very colorful.

For example: Applied Animal Behaviour Science 168 (2015) 1823.
Asian-Australasian Journal of Animal Sciences (AJAS) 2017; 30(2):
267-274.

Published, perhaps not included, there are many studies related to EE in the Chinese and Japanese literature that suggest removing Fig3 and distinguishing the southern hemisphere from the northern hemisphere means little for talking about "Global overview"

Response: thank you for your response. Papers were included based on the key-words used. The above-mentioned papers does not include those key-words. There are certainly studies that do not use the words enrichment and environmental in their titles or keywords. To try to get round this, we checked all the reference lists of the included articles. Even so, there is a chance that articles will be left out of the search. We have therefore inserted a sentence at the end of the discussion drawing attention to this.

Reviewer 2 Report

Comments and Suggestions for Authors

As critical comments, I would like to highlight the following recommendations:

1) it is desirable to avoid annoying repetitions of information, as the authors too often repeat essentially the same information in the discussion of the results, as well as in the conclusion and conclusions;

2) it would be desirable to identify and show the «root» publications on which the analyzed thousands of publications were mainly based, in order to better present the dialectic of such study.

Author Response

REVIEWER 2

As critical comments, I would like to highlight the following recommendations:
1) it is desirable to avoid annoying repetitions of information, as the authors too often repeat essentially the same information in the discussion of the results, as well as in the conclusion and conclusions;

Response: we modified the text of the Discussion section to try to avoid the cited repetitions. Modifications are marked in red.

2) it would be desirable to identify and show the «root» publications on which the analyzed thousands of publications were mainly based, in order to better present the dialectic of such study.

Response: all the publications analysed can be seen in the Raw Data file (see the Data Availability Statement section). We inserted a sentence informing this on the paper (in red).

Best wishes, Rob